# Clinical Grade Human Pluripotent Stem Cell-Derived Engineered Skin Substitutes Promote Keratinocytes Wound Closure In Vitro

**DOI:** 10.3390/cells11071151

**Published:** 2022-03-29

**Authors:** Sophie Domingues, Annabelle Darle, Yolande Masson, Manoubia Saidani, Emilie Lagoutte, Ana Bejanariu, Julien Coutier, Raif Eren Ayata, Marielle Bouschbacher, Marc Peschanski, Gilles Lemaitre, Christine Baldeschi

**Affiliations:** 1Centre d’Etude des Cellules Souches, 91100 Corbeil-Essonnes, France; sdomingues@istem.fr (S.D.); adarle@istem.fr (A.D.); yolande.masson@club-internet.fr (Y.M.); msaidani@istem.fr (M.S.); abejanariu@istem.fr (A.B.); jcoutier@istem.fr (J.C.); mpeschanski@istem.fr (M.P.); 2URGO RID, 42 Rue de Longvic, 21300 Chenôve, France; lagoutte_emilie@yahoo.fr (E.L.); m.bouschbacher@fr.urgo.com (M.B.); 3INSERM U861, I-Stem, AFM, Institute for Stem Cell Therapy and Exploration of Monogenic Diseases, 91100 Corbeil-Essonnes, France; rayata@istem.fr (R.E.A.); gilles.lemaitre@univ-evry.fr (G.L.); 4Université Paris-Saclay, Université d’Evry, U861, 91100 Corbeil-Essonnes, France

**Keywords:** pluripotent stem cells, skin tissue engineering, keratinocytes, fibroblasts, wound healing, GMP compliant

## Abstract

Chronic wounds, such as leg ulcers associated with sickle cell disease, occur as a consequence of a prolonged inflammatory phase during the healing process. They are extremely hard to heal and persist as a significant health care problem due to the absence of effective treatment and the uprising number of patients. Indeed, there is a critical need to develop novel cell- and tissue-based therapies to treat these chronic wounds. Development in skin engineering leads to a small catalogue of available substitutes manufactured in Good Manufacturing Practices compliant (GMPc) conditions. Those substitutes are produced using primary cells that could limit their use due to restricted sourcing. Here, we propose GMPc protocols to produce functional populations of keratinocytes and fibroblasts derived from pluripotent stem cells to reconstruct the associated dermo-epidermal substitute with plasma-based fibrin matrix. In addition, this manufactured composite skin is biologically active and enhances in vitro wounding of keratinocytes. The proposed composite skin opens new perspectives for skin replacement using allogeneic substitute.

## 1. Introduction

Skin wound healing is a complex biological process involving a huge variety of cells producing matrix modeling proteins, proteinases, cytokines such as chemoattractants, growth factors, and angiogenic factors [1]. Chronic wounds, such as leg ulcers associated with sickle cell disease, pressure ulcers, and vascular ulcers, occur as consequence of a prolonged inflammatory phase during the healing process [2,3,4]. They are extremely hard to heal and persist as a significant health care problem due to the absence of effective treatment and the uprising number of patients [5]. Development and implementation of wound management strategies that focus on increasing health-related quality of life and effectively reduce costs for patients and healthcare systems are needed.

Current treatment for chronic wounds involves preventing wound infections, debriding the tissue and selecting appropriate dressings to maintain favorable wound-healing environment. Many wound dressings have been developed to try to both protect the healing wound from infection and also to help promote the wound healing process itself [1,6,7]. As a matter of fact, the approach of regenerative medicine using stem cells has emerged to provide new therapeutic options in the domain of wound healing [8].

The most common stem cells used in skin regeneration and wound healing are adult stem cells owing to containing significant proliferative capacity and having the ability to differentiate into limited tissue-specific cells. Among the different types of adult stem cells, keratinocytes (KER) or mesenchymal stem cells (MSCs) have gained considerable attention as suitable candidates to enhance tissue regeneration [1,7,8,9].

Besides adult stem cells, human pluripotent stem cells (hPSCs), such as embryonic stem cells (hESCs) and induced pluripotent stem cells (hiPSCs), are able to self-renew and differentiate into all three germ layers of the embryo—ectoderm, mesoderm, and endoderm [10]. Their proliferative and differentiation capacities are highly convenient for cell substitution therapy because they enable the propagation of cells to obtain the required amounts and the possibility of creating any cell type from the human body. Since 2009, our group and others have developed research grade protocols allowing the differentiation of hPSCs into various type of cells, including major populations present in skin: keratinocytes [11,12,13], fibroblasts [14,15], melanocytes, and dendritic cells [16] to engineer skin substitutes.

In order to use hPSCs for clinical use, it is necessary to manufacture and perform quality control of hPSCs according to current Good Manufacturing Practices (GMP) such as those defined by the European Medicines Agency (EMA, https://www.ema.europa.eu/en, accessed on 1 September 2021) in the European Union and by the US Food and Drug Administration (FDA, https://www.fda.gov/, accessed on 1 September 2021) in the USA. Briefly, GMPc requires that medicines (a) are of consistent high quality; (b) are appropriate for their intended use; and (c) meet the requirements of the marketing authorization or clinical trial authorization. The establishment of hPSC lines must be carried out in accordance with the relevant laws and policies of the country where the derivation is performed. Information on the current legal position, ethical and regulatory oversight for EU countries can be found on https://www.eurostemcell.org (accessed on 1 September 2021). However, there are directives and guidelines in force for all EU countries that specify good manufacturing practices relevant for the establishment of hPSCs that must be obeyed. By the end of 2019, there were reports on at least 54 clinical studies based on hPSCs [17]. Nearly half of these studies aim to develop new therapies for eye diseases, mainly for different kinds of macular degeneration and retinal dystrophies. No current clinical trial is related to skin healing and it will be essential to fill this cavity.

This work focused on the development of a new composite-engineered skin produced under clinically compatible conditions to provide a biological dressing for wound healing using unlimited sources of cells obtained by differentiating hPSCs. Keratinocytes and fibroblasts derived from hPSCs were generated under GMPc and large-scale culture conditions. The resulting cells, when co-cultured on a human fibrin matrix, showed regenerative potential in a keratinocyte-based scratch wound assay.

## 2. Materials and Methods

### 2.1. hPSC Culture

hESC source: Clinical grade hESC line RC-9 derivation was previously described by Roslin Cells Laboratory [18]. Human iPSC source: PC-1432 line was reprogrammed by Phenocell (Grasse, France), using OSKM episomal technics. Informed consent was obtained according to the ethical guidelines and the French regulatory legislation. hPSC lines were cultured on L7 Matrix at 10 μg/mL (Lonza Bioscience, Morrisville, NC, USA) and maintained by media renewal using StemPro^®^ hES SFM medium (Thermo Fisher Scientific, Waltham, MA, USA) supplemented with stabilized FGF2 at 10 ng/mL (Miltenyi Biotec, Bergisch Gladbach, Germany) every two days and manually passaged each week as small clumps.

### 2.2. Karyotype Analysis

Karyotype analysis was performed on hPSC and their derivatives (KER and FIB) by m-FISH analysis. For hPSC and FIB, non-confluent cells were treated in their respective media with colchicine at 20 µg/mL (Eurobio, Les Ulis, France) for 1h30 before hypotonic choc with 5.6 mg/mL of KCl (Sigma Aldrich, St. Louis, MI, USA) and fixation. For KER, cells were cultured in their medium and treated daily with isoboost 50x solution (CELLnTEC) for 3 days to synchronize the cells. At day 3, cells were treated with colchicine at 20 µg/mL for 24 h before hypotonic choc in a mix of medium and water (1:3) and fixation was proceeded. Analysis was carried out by m-FISH staining using mFISH 24Xcite probe (MetaSystems, s.r.l., Milan, Italy), according to the manufacturer’s instructions. The metaphase’s acquisitions were carried out with an AxioImager Z2 microscope equipped with a camera cool cube and 10x and 63x plan apo objectives (Zeiss, Oberkochen, Germany) piloted by Metafer4 software (version 3.11.8, Metasystems, s.r.l., Milan, Italy). The karyotypes (30 to 50 metaphases per analysis) were then analyzed and classified using Isis software (version 5.7.8, Metasystems, s.r.l., Milan, Italy).

### 2.3. Immunofluorescence Analysis

For immunocytochemistry, hPSC, KER, and FIB were fixed in 4% paraformaldehyde (Euromedex) and permeabilized with 0.1% Triton X-100 (Sigma, St. Louis, MI, USA). After blocking non-specific interactions with 5% bovine serum albumin (Sigma-Aldrich), samples were incubated with primary antibodies overnight at +4 °C in blocking buffer. After washing, samples were incubated 1 h with species-specific fluorophore-conjugated secondary antibodies and counterstained with DAPI at 1 μg/mL (Millipore) to allow nuclei detection. Image acquisition was performed with an inverted microscope (Axio Imager, Zeiss, Oberkochen, Germany) or the HighContentScreening module CellInsight CX7 (Thermo Fisher Scientific, Waltham, MA, USA), after cell segmentation and thresholding was based on negative control (secondary antibody only).

For immunohistochemistry and Hematoxylin Eosin staining (HE), tissues were fixed in 10% formaline (VWR) before paraffin embedding. All IHC staining was carried out on paraffin sections of 4 µm thickness using Ventana Discovery XT IHC module according to the manufacturer’s instructions. HE image acquisition was performed with EVOS™ XL Core Imaging System (Thermo Fisher Scientific). IHC image acquisition was performed with an inverted microscope (Axio Imager, Zeiss). The list of antibodies is presented in Appendix A.

### 2.4. Flow Cytometry Analysis

KER and FIB were harvested using Trypsin-EDTA (Invitrogen, Waltham, MA, USA) and hPSCs were harvested using accutase (Life Technology, Carlsbad, CA, USA). BD Cytofix/Cytoperm™ Fixation/Permeablization Kit (BD Biosciences, Franklin Lakes, NJ, USA) was used to fix cells and for antibody incubation according to the manufacturer’s instructions. All cell types were incubated for 30 min at room temperature with fluorochrome-conjugated primary antibodies (Appendix A), washed and processed for flow-cytometry analysis. All experiments were performed with MACSQuant system (Miltenyi Biotec) on 20,000 events per sample and analyzed using Flowjo software. Acquisition parameters were set up on a control cell line, and maintained for all analyzed cell lines. The list of antibodies is presented in Appendix A.

### 2.5. RT-qPCR

Total RNA was extracted with the RNeasy Mini extraction kit (Qiagen, Hilden, Germany) using QIAcube instrument and mRNA reverse transcription was performed using Superscript III kit on nexus GSX1 thermocycler (Eppendorf, Hamburg, Germany) according to the manufacturer’s protocols. Samples were analyzed using a custom TaqMan Gene Expression Array Plate (4413260, Thermo Fisher Scientific, Waltham, MA, USA) and plates were read with QuantStudio™ 7 Flex instrument according to the manufacturer’s instructions. The list of genes and associated Taqman probes are presented in Appendix A. Quantification of gene expression was based on the Delta Ct method, normalized with housekeeping genes expression.

### 2.6. Keratinocyte Differentiation and Culture

After amplification, hPSCs were passaged and seeded in new L7 matrix-coated flask (Nunc™ EasYFlask™, 225 cm^2^, Thermo Fisher Scientific) at 1 clump per cm^2^ with StemPro^®^ hES SFM medium supplemented with stabilized FGF2 for one day (D0). The medium was replaced by D-KSFM^®^ (defined Keratinocytes Serum-Free Medium, Invitrogen, Waltham, MA, USA) supplemented with 0.273 nM BMP-4 (Peprotech, Cranbury, NJ, USA) and 1 μM retinoic acid (Sigma Aldrich) on day 1 and 3 to induce differentiation. On D6, medium was switched to D-KSFM^®^ alone until the end of differentiation stage (between D15 and 21). Keratinocytes derived from hPSC (KER-hPSC) were sorted by differential trypsinization (0,05% *v*/*v*; Invitrogen) and amplified in higher certified KER medium (CnT-07.HC; CELLnTEC) on medical grade collagen type I (Collagen Solutions, Eden Prairie, MN, USA) coated CellSTACK^®^ 5-chamber (Corning^®^, Corning, NY, USA, 3180 cm^2^) at 30,000 cells per cm^2^ until 100% confluency during 2 passages for the maturation stage of the process. Cells were then frozen in an animal-component-free, defined cryopreservation medium with 10% DMSO (Cryostor CS10, Biolife Solutions, Bothell, DC, USA) at the end of passage p1 (Figure 1). All quality controls were carried out after thawing of KER-hPSC at p2.

### 2.7. Fibroblast and Myofibroblast Differentiation and Culture

After amplification, hPSCs were passaged and seeded in a new L7 matrix-coated flask (225 cm^2^) at 1 clump per cm^2^ with StemPro^®^ hES SFM medium supplemented with stabilized FGF2 for one day (D0). The medium was replaced by fibroblast medium (CnT-PR-F; CELLnTEC, Bern, Switzerland) supplemented with 5% final concentration of defined and irradiated FBS (Hyclone, Cytiva Lifesciences, Marlborough, MA, USA) and 0.273 nM BMP-4 on day 1 and 3 to induce differentiation. On D7, the medium was switched to fibroblast medium supplemented with 5% FBS until the end of differentiation part (D14). Fibroblasts derived from hPSC (FIB-hPSC) were sorted by trypsinization and amplified on new L7 matrix-coated flasks with fibroblast medium supplemented with 5% FBS at 50,000 cells per cm^2^ until the end of p0. At p1, FIB-hPSCs were passaged and amplified without coating and with fibroblast medium supplemented with 5% FBS at 20,000 cells per cm^2^. Cells were then frozen in Cryostor CS10 at the end of passage p2 (Figure 2). All quality controls were carried out after thawing of FIB-hPSC at p3.

For myofibroblast differentiation, the FIB-hPSCs at passage 3 were seeded into collagen type I coating wells at 10,000 cells per cm^2^ in fibroblast medium supplemented with 5% FBS. To initiate myofibroblast differentiation, cells were treated with 10 ng/mL TGF-β1 (Peprotech) for 4 days before analysis.

### 2.8. 3D Cell Culture and Analysis

Organotypic epidermis were generated on polycarbonate culture inserts (Millipore, Burlington, MA, USA), using KER-hPSC (passage 2) or primary keratinocytes (HPEK, Appendix A) maintained in CnT-07.HC medium in immersion for 48 h to allow the cell attachment on the membrane at 500,000 cells per cm^2^. The medium was switched to a reconstruction medium (CnT-PR-3D, CELLnTEC) for 24 h. Finally, KER-hPSC or HPEK were placed at the air-liquid interface for 21 days, to allow stratification. Medium was renewed every two days.

For dermo-epidermal substitute’s production, dermal equivalents were generated with fibrin matrix populated with primary fibroblasts (HDFN, Appendix A) or FIB-hPSC at 10,000 cells per cm^2^, on polycarbonate culture inserts. The plasma (EFS-ABG) was mixed with a saline solution composed of sodium chloride, calcium chloride, and Exacyl (Sanofi) all from GMPc providers. The prepared fibrin was spread out in a culture insert and put at +37°C during 1–2 h to permit the coagulation of the matrix. After coagulation, the dermal equivalents were cultured for 6 days with ECM medium (CnT-PR-ECM, CELLnTEC) in immersion phase (until D7). KER-hPSC or HPEK were seeded on the dermal equivalent or on fibrin matrix alone at 100,000 cells per cm^2^ and cultured for 7 days (until D14) in CnT-07.HC medium supplemented with 400µg/mL of Exacyl^®^ (Sanofi, Paris, France). For the dermo-epidermal reconstruction, medium was switched to a reconstruction medium (CnT-PR-FTAL5, CELLnTEC) for 24 h (D15). Finally, tissues were placed at the air-liquid interface in the same medium for 21 days, to allow stratification. Media were renewed every two days (until D36).

Epidermis thickness was measured using the “line” module of the Axio Imager software (Zeiss, Oberkochen, Germany). For each type of dermo-epidermal tissue, 6 measures per image (5 images per replicate and 3 replicate per condition) were made between the beginning of the basal layer and the beginning of the corneal layer.

Assessment of keratinocyte polarity was performed in reconstructed epidermis as previously described [19]. Briefly, regions of interest (ROI) were defined in tissue sections stained with DAPI, corresponding to ~600 µm section length (20 images per condition). A mask was defined by the experimenter to extract the basal keratinocyte layer and characterize nuclei orientation versus the dermo-epidermal junction (DEJ). Angle measurements were performed automatically using a routine developed with the Fiji software (version: v1.53c, University of Wisconsin-Madison, USA), and data were plotted into angle categories (from 0° to 90°) using the R software (Genethon imagery platform, Evry, France).

### 2.9. Scratch Wound Assay and Viability Analysis

Keratinocytes (HPEK) used in this study are presented in Appendix A. Keratinocytes were seeded at 30,000 cells per well in an ImageLock TM^®^ 96-well plate (EssenBioScience, Ann Arbor, MI, USA) and cultured for 24 h. At T = 0 h, one part of the plates with keratinocytes monolayers were scratched using Incucyte^®^ 96-well WoundMaker Tool (EssenBioScience) and treated with conditioned media from the different reconstituted tissues. The plates were incubated 160 h at 37 °C, 5% CO_2_ in the IncuCyte Zoom system. Pictures of entire wells were taken every 2 h for 160 h, using IncuCyte ZOOM™ live-cell imaging system (EssenBioScience). The analysis algorithm automatically threshold to each image to identify the position of the wounded and unwounded zones, to deliver measurements of wound size for the entire time-course of the experiment. The other part of the plates was treated with conditioned media from the different reconstituted tissues without scratching to evaluate the cytotoxicity. Cells viability was measured with IncuCyte S3™ live-cell imaging system (EssenBioScience) according to the manufacturer’s instruction. Propidium iodide (Thermofisher Scientific) was used to identify apoptotic cells. The analysis algorithm automatically threshold to each image to identify the total cell area and propidium iodide positive-cell area of each well to deliver measurements of viability for the three selected time-course of the experiment (T = 0 h, T = 80 h and T = 160 h).

### 2.10. Mycoplasma Detection

The mycoplasma presence was verified during all processes of production for each batch of KER, FIB, and reconstructed tissues with MycoAlert Mycoplasma Detection Kit (LT07-318, Lonza^®^, Basel, Germany) according to the manufacturer’s instructions.

### 2.11. Statistical Analysis

Statistical analyses were performed using Mann–Whitney nonparametric *t*-test for side-by-side comparison (GraphPad Prism 8.2.1). Results are expressed as mean ± SD. *p* values < 0.05 were considered significant. For statistical significance, ns = non-significant, * *p* < 0.05; ** *p* < 0.01; *** *p* < 0.005; **** *p* < 0.001.

## 3. Results

### 3.1. Characterization of Compatible Clinical Grade hESC

Before differentiation steps, clinical grade hESC RC-9 line and research grade hiPSC PC1432 line were cultivated and controlled in large-scale GMPc condition in order to have a unique bank for the production of KER and FIB.

After 2 weeks of amplification, the two cell lines were frozen with GMPc freezing medium to generate hPSC banks of about 150 to 200 cryovials (500,000 to 1,000,000 of cells in clumps format per cryovials) per cell line. Both cell lines were then characterized to ensure their quality. The karyotype of each cell line (Appendix A) was verified and no abnormalities were found (46; XY). Concerning the phenotype, all hPSCs expressed the pluripotency markers Oct-4, Nanog, SSEA-4, and TRA-1-60 (Appendix A). The quantification of Oct-4, Nanog, SSEA-4, and TRA-1-81 by flow cytometry revealed that more than 90% of the cells were all positives (Appendix A).

To evaluate the pluripotency capacity of the hPSC lines, embryoid bodies (EB) were generated (Appendix A) and analyzed. After 14 days of culture, EB from each cell line no longer expressed pluripotency marker genes *SOX2*, *POU5F1*, *NANOG*, and *LIN28* (Appendix A). The hPSC-derived EB presented an expression of mesodermal markers *MYF5, MYOD1, GATA4,* and *CDX2* ((Appendix A), ectodermal markers *KRT1*, *OLIG2*, *NODAL*, and *KRT19* (Appendix A), and endodermal markers *AFP*, *GATA6*, *PDX1*, and *CXCR4* (Appendix A).

### 3.2. Development of Keratinocyte Differentiation Protocol Compatible with Clinical Grade Regulatory Standards

Next, we evaluated the potential of the clinical hESC line to give rise to a homogeneous and functional population of keratinocytes (KER). Based on a previously described protocol [12], a defined and clinically compatible differentiation protocol was established (Figure 1a).

All used components were examined by a quality control manager to be in accordance with GMPc guidelines. Briefly, hESC were seeded and mass cultured in 225 cm^2^ culture L7 matrix-coated flasks (5625 cm^2^ in total) for approximatively 15 days in defined conditions for KER differentiation. Then, KER were amplified into six large-scale CellSTACK^®^ 5-chamber culture containers (19.080 cm^2^ in total) for two passages before banking. At this step, the KER population was composed of 1.5 billion of cells and was frozen in order to have at least 300 cryovials. After banking, KER were thawed to check their phenotype and functionality (at p2). This quantity of KER allows the production of approximately 15,000 cm^2^ of epidermal substitute.

Microscopy analysis of these cells showed a morphology with highly packed cobblestone shaped cells, a high nucleus-to-cytoplasm ratio, a morphology typical of basal keratinocyte similar to human normal primary culture “HPEK” (Figure 1b).

The obtained KER presented basal keratinocyte markers keratin 5 and keratin 14 similarly to HPEK (Figure 1b). The cell population revealed large expression of keratin 19, whereas in the HPEK, only a few cells expressed this marker (Figure 1c). The quantification of those markers by flow cytometry indicated that the KER-hESC and HPEK populations were composed with more than 99% of keratin 5 and keratin 14 positive cells (Appendix A). KER-hESC population presented 99.3% keratin 19 positive cells, whereas only 12.4% for HPEK population (Appendix A).

KER population was also composed with more than 99% of p63, α6 integrin, and β4 integrin positive cells, similarly to HPEK (Figure 1c). The proliferative capacity of KER-hESC was also measured (Figure 1d). First, cell doubling time was calculated after 7 days of culture and revealed a similar capacity of KER-hESC at passage 2 and HPEK at passage 3 to expand with less than 50 h of doubling time (Figure 1d). KER-hESC were able to proliferate until at least passage 4. KER-hESC karyotype was normal after the differentiation process (Figure 1e). To evaluate the functionality of KER-hESC, in vitro epidermal reconstitution was performed on polycarbonate insert (Figure 1f). Reconstructed epidermises were quite similar between KER-hESC and HPEK (Figure 1f), with the presence of all specific layers commonly present in human skin (basal, spinous, granulous, and corneal layers).

### 3.3. Development of Fibroblast Differentiation Protocol Compatible with Clinical Grade Regulatory Standards

We developed a process to differentiate the clinical hESC line to a homogeneous and functional population of FIB. Based on a previously published protocol [14], we established a defined and clinically compatible differentiation protocol (Figure 2a).

All used components were examined by a quality control manager to be in accordance with GMPc guidelines. Briefly, the hESC line was seeded and cultured for approximatively 14 days in five culture L7 matrix-coated 225 cm^2^ flasks (1125 cm^2^ in total) in defined conditions. At the end of differentiation, the FIB-hESC population was mass cultured (4500 cm^2^ in total) to allow cell maturation before banking at the end of passage 2. The FIB-hESC pool was composed of 300 million of cells and frozen in Cryostor in order to have at least 60 cryovials. After banking, FIB-hESC were thawed to check their phenotype and functionality (at p3). This amount of FIB allows the production of approximately 15 000 cm^2^ of dermal tissue.

Microscopy analysis of these cells showed a similar morphology to normal human dermal fibroblasts neonatal (HDFN). Indeed, they were large, flat, and elongated (Figure 2b). The obtained population expressed the fibroblast markers serpin H1 and fibronectin such as HDFN (Figure 2b). The quantification of mesenchymal markers CD73 and CD166 by flow cytometry showed the presence of more than 95% positive cells in FIB-hESC as in HDFN (Appendix A). The quantification of FIB markers by flow cytometry presented an expression with more than 85% of FAP (Fibroblast Activation Protein) positive cells and more than 95% of vimentin positive cells similar to HDFN (Figure 2c). Cell populations were also composed of more than 90% of podoplanin (papillary dermal fibroblast marker) positive cells, similar to HDFN (Figure 2c). We also measured the proliferative capacity of FIB-hESC. First, the doubling time was calculated after 4 days of culture and revealed a faster proliferation capacity of FIB-hESC at passage 3 with 47 h compared to HDFN at passage 4 with 101 h of doubling time (Figure 2d). The doubling time of FIB-hESC after p3 increased with passages to reach a similar doubling time to HDFN after passage 5. The cell karyotype was normal after the differentiation process (Figure 2e). The capacity of fibroblasts to differentiate into myofibroblasts, implicated in wound healing process, was evaluated by treatment of these cells with TGF-β1 for 4 days. The presence of the myofibroblast marker αSMA, a marker of contractile activated fibroblast, was then analyzed and quantified (Figure 2f,g). In absence of TGF-β1 treatment, no αSMA staining was detected for HDFN and FIB-hESC. After TGF-β1 treatment, 1.8% of HDFN and 2.6% of FIB-hESC were αSMA positives (Figure 2g).

### 3.4. Characterization of Dermo-Epidermal Tissue Using GMPc Plasma-Based Fibrin Matrix

After the production and the characterization of the two cell types involved in the future skin engineered tissue, FIB-hESC were mixed with a fibrin gel to produce a dermal equivalent (Figure 3a). After a short period of culture allowing dermal modelling by the cells, KER-hESC were seeded on top of that compartment and cultured until a confluent monolayer was observed.

The dermo-epidermal tissue was then placed at air liquid interface several weeks to evaluate its capacity to reconstruct a pluristratified epidermis. The composite-engineered skin containing KER and FIB derived from hESC presented a similar stratification to HPEK and HDFN dermo-epidermal reconstituted tissue (Figure 3b).

These engineered skins presented the fibroblast marker vimentin in the dermal compartment, the keratin 5 marker at the basal epidermal layer, and the suprabasal markers involucrin and loricrin at the upper location as for primary cells tissues (Figure 3b). The separation between the dermal and the epidermal part was all defined and considered as the dermo-epidermal junction (DEJ).

Thickness measurement was performed between the JED and the beginning of the corneal layer and showed a difference between the primary cells and the hESC-derived dermo-epidermal tissues (Figure 3c). [HPEK + HDFN] epidermis measured 60 µm of height, whereas [KER + FIB]-hESC epidermis was thicker with 105 µm.

Measurements of basal nuclei orientation were performed on each dermo-epidermalreconstructed tissue sections by image analysis of stained nuclei (Figure 3d). In healthy skin, basal keratinocytes are naturally oriented perpendicularly to the dermo-epidermal junction. Nuclei orientations versus the DEJ plane were determined and classified according to three categories: nearly perpendicular (angles between 60° and 90°), nearly parallel (angles between 0° and 30°), and oblique (angles between 30° and 60°). In HPEK + HDFN dermo-epidermal reconstituted tissues, two groups of nuclei were mostly present, nearly oblique and nearly parallel orientations, whereas in [FIB + KER]-hESC, the majority of nuclei were from oblique to nearly perpendicular (Figure 3d).

### 3.5. Conditioned Media from GMPc Human Embryonic Stem Cell-Derived Engineered Skin Substitutes Improve Keratinocytes Wound Closure In Vitro

Wound healing capacity of the conditioned media of our skin substitutes was evaluated by performing a scratch wound assay in 2D culture on keratinocyte monolayers.

The conditioned media collected from reconstructed tissues with fibrin alone, FIB-hESC alone, or KER-hESC alone, and finally, from the composite-engineered skin [FIB + KER]-hESC (Figure 4a) were applied on an unscratched and a scratched KER confluent layer.

First, the viability of unwounded primary keratinocytes was measured for 160 h to evaluate a potential toxic effect of these conditioned media on the monolayer (Figure 4b). The results showed that under all conditions, there was no major impact on viability after 80 h and 160 h, with more than 92% of living cells in the wells. Next, these conditioned media were applied on scratched keratinocyte monolayers (Figure 4c). The conditioned medium collected from the fibrin alone tissue showed a negative impact on the scratched KER monolayer: the wound was not closed but on the contrary, it had widened and no more cells were visible in the field after 160 h (Figure 4c). With conditioned media from the FIB-hESC alone or KER-hESC alone tissues, wounds were also not closed but cells around the wound were still present (Figure 4c).

The wound was completely healed with the medium obtained with [FIB + KER] tissue (Figure 4c). These observations were confirmed with the analysis of the wound size evolution for the 160 h of the experiment (Figure 4d). After 160 h, the wound size was out of 2000 µm for fibrin alone condition and near the initial scratch (850–550 µm) for the FIB-hESC alone or KER-hESC alone media. Treatment using conditioned medium obtained with [FIB + KER] tissue allowed a wound closure near 130 h (Figure 4d). The size of the four wounds was compared at 80 h after the beginning of the conditioned medium application (Figure 4e).

At that time, the wound size was at 1400 µm for fibrin alone, at 850–900 µm for FIB-hESC alone, 600 µm for KER-hESC, and less than 400 µm for the composite-engineered skin conditioned medium (Figure 4e). This effect was amplified all along the kinetic until 160 h (Figure 4c) and a total wound closure was obtained only with the conditioned medium collected from the composite-engineered skin (Figure 4d).

The same observations were made with primary cells (Appendix A). The viability of the unwounded keratinocyte monolayer was greater than 93% after 80 h and 160 h of application of the conditioned media (Appendix A). The wound size was closed to the initial size (850–550 µm) with HDFN alone or HPEK alone tissues media. Using conditioned medium obtained with [HDFN + HPEK] tissue, the wound was closed after 140 h of culture (Appendix A). At 80 h the size of the three wounds was compared (Appendix A) and was at 800 µm for HDFN alone, 600 µm for HPEK, and less than 200 µm for the composite-engineered skin conditioned medium (Appendix A).

### 3.6. Evaluation of GMPc Protocols Using Human-Induced Pluripotent Stem Cells

Same experiments were carried out using hiPSC in order to confirm the robustness of the data obtained with hESC. KER-hiPSC population was composed of more than 99% of keratin 5 and keratin 14, and 92% for keratin 19 positive cells. This cell population presented more than 98% of p63, α6, and β4 integrin positive cells (Appendix A).

Microscopy analysis of these cells showed a morphology with highly packed cobblestone shaped cells and a high nucleus-to-cytoplasm ratio. The obtained cells presented expression of keratin 5, keratin 14, and keratin 19 (Appendix A), similarly to KER-hESC by immunostaining. KER-hiPSC at passage 2 had a proliferative profile similar to KER-hESC with less than 60 h of doubling time (Appendix A). KER-hiPSC karyotype was normal after the differentiation process (Appendix A) and the in vitro epidermal reconstitution was similar to KER-hESC (Appendix A), with the presence of all specific layers commonly present in human skin.

FIB-hiPSC population was composed of 98% CD73 and CD166 positive cells, more than 84% of FAP positive cells, and more than 99% of vimentin positive cells (Appendix A), similar to FIB-hESC. Microscopy analysis of these cells showed a similar morphology to normal human dermal fibroblasts neonatal (HDFN). Indeed, these cells were large, flat, and elongated. The obtained population expressed the fibroblast markers serpin H1 and fibronectin (Appendix A). The proliferation capacity of FIB-hESC was measured at passage 3 with 41 h of doubling time (Appendix A). The cell karyotype was normal after the differentiation process (Appendix A).

FIB-hiPSC were differentiated into myofibroblasts with TGF-β1 treatment for 4 days. The presence of the myofibroblast marker αSMA was analyzed and quantified (Appendix A). In the absence of TGF-β1 treatment, no αSMA staining was detected as for HDFN and FIB-hESC. After TGF-β1 treatment, 2.2% of FIB-hiPSC were αSMA positives (Appendix A).

The composite-engineered skin containing KER and FIB derived from hiPSC showed a similar stratification compared to hESC-derived dermo-epidermal reconstituted tissue (Appendix A). These engineered skins presented the fibroblast marker vimentin in the dermal compartment, the keratin 5 marker at the basal epidermal layer, and the suprabasal markers involucrin and loricrin at the upper location as for primary cell tissues.

Thickness measurement showed a difference between the primary cells and the hiPSC-derived dermo-epidermal tissues (Appendix A). [HPEK + HDFN] epidermis measured 60 µm of height, whereas [KER + FIB]-hiPSC epidermis was similar with 66 µm. Nuclei orientations versus the DEJ plane were determined. In [FIB + KER]-hiPSC, the majority of nuclei were oblique (Appendix A).

Wound healing capacity of the conditioned media of hiPSC-derived skin substitutes was evaluated by performing scratch wound assay in 2D culture on keratinocyte monolayers. The viability of the unwounded keratinocyte monolayer was also more than 93% after 80 h and 160 h after conditioned media application (Appendix A). The conditioned media collected from reconstructed tissues with FIB-hiPSC alone or KER-hiPSC alone or from the composite-engineered skin [FIB + KER]-hiPSC (Appendix A) were applied on a scratched KER confluent layer. FIB-hiPSC alone or KER-hiPSC alone tissues conditioned media do not stimulate wound closure, whereas carried out with [FIB + KER] tissue conditioned medium (Appendix A). The wound size was around the initial size (600–1000 µm) with FIB-hiPSC or KER-hiPSC alone tissues conditioned media. Concerning conditioned medium obtained with [HDFN + HPEK] tissue, the wound was closed after 145 h of culture (Appendix A). At 80 h the size of the three wounds was compared and was at 600 µm for FIB-hiPSC alone, 600 µm for KER-hiPSC, and less than 400 µm for the composite-engineered skin conditioned medium (Appendix A).

All these data are in complete correlation with those obtained using hESC confirming the relevance of this GMPc protocol whatever the cell sourcing.

## 4. Discussion

In this study, we developed clinical grade compatible processes to produce a composite-engineered skin elaborated with hPSC-derived KER and FIB, complexed with a human fibrin scaffold. These cells were able to generate a pluristratified epidermis on the reconstructed dermal compartment proving the potency of these raw cellular products. This way, we are able to propose a biological dressing that could be used for further clinical trials. These processes were developed using the clinical compatible grade human embryonic cell line, RC-9 [18] and assayed with a research grade human iPSC line in order to validate and improve the robustness of the process whatever the type of hPSC source.

In addition, compared to previously described protocols, these processes were performed entirely under culture conditions compatible with clinical use, from amplification of hPSC to banking of differentiated cells (Appendix A).

As several recent protocols for KER and FIB differentiation, hPSC were cultured without feeder cells and induction of differentiation was carried out using recombinant proteins. However, a sum of these protocols were previously developed in semi-defined media containing bovine serum or using undefined-coated culture dishes (such as Matrigel or Geltrex) but were not totally compliant with clinical use (see Appendix A) [14,20,21,22,23,24]. In this study, all the components used whether for culture plastic coating or media are animal/xeno-free raw materials. Moreover, for some specific components such as FBS all the manufacturer documentation enabling to respond to regulatory agencies guidelines were obtained. The amplification and storage of cells were also performed using defined media and GMPc cryopreserved solution.

Moreover, protocols developed in this study allowed us to obtain pure populations of KER and FIB in large-scale productions while using hPSC colony clumps, thus showing efficiency of our differentiation processes.

Phenotypic characterization of obtained cells indicates that hPSC-derived keratinocytes present similar markers to normal adult ones, except for keratin 19. Indeed, more than 90% of hPSC-derived keratinocytes express K19 protein compared to 12.4% for normal adult keratinocytes (Appendix A). This specific keratin is considered as a potential biochemical marker of juvenile skin cells in vivo and in vitro [23]. Moreover, the FIB-hPSC population was composed of more than 90% of podoplanin positive cells, a specific dermal papillary FIB marker similar to primary cells [25,26]. As these two biological drug substances (KER-hPSC and FIB-hESC) harbored these types of markers, the generated skin substitutes could have at least the same properties as composite-engineered skin produced with juvenile primary cells. This type of “juvenile-like” product could be more appropriate in a clinical perspective since aged skin failed to heal after grafting due to a decrease in proliferative potential [27].

The composite 3D structure is based on human fibrin scaffold seeded with FIB-hPSC layered by KER-hPSC. The fibrin matrix is actually one of the gold standards in skin cellular therapy and was used previously in clinical trials [28,29]. Moreover, in this study, all batches of plasma used to prepare the fibrin matrix were manufactured according to French and European regulatory agencies guidelines. That manufacturing process included a step with the pooling of several donors’ plasma in order to have a homogeneous fibrin product suitable for the production of a large number of reconstituted tissues, and to limit plasma’s donor impact.

In order to propose this product for a human clinical trial, we demonstrated that cells derived from hPSC are able to form a pluristratified epidermis on a dermal equivalent scaffold (Figure 3c and Appendix A). In addition, we performed a 2D scratch wound healing using keratinocytes, a relevant assay for therapeutic activity that could be used as a potency test in quality control of the drug product. Conditioned medium of the 3D composite-engineered skin was assessed on a scratch assay on normal keratinocytes culture. Results obtained in this study confirmed that the presence of the two types of cells (fibroblasts and keratinocytes) is indispensable to promote a better healing. Several studies have proved the important role of dermal fibroblasts (FIB) in skin remodeling and wound healing [30]. They can produce extracellular matrix (ECM) components such as collagen and fibronectin, and can stimulate keratinocyte growth and differentiation by either secreting cytokines and/or growth factors or via cell–cell and cell-ECM contacts. In turn, KER can positively affect fibroblasts proliferation [31,32]. Therefore, the combination of human KER and FIB, in a composite skin substitute is able to implement dermo-epidermal structure, but also to deliver some growth factors such as epithelial growth factor (EGF) or vascular endothelial growth factor (VEGF) and extracellular matrix that could facilitated wound healing. These composite-engineered skins, with primary cells or derived from hPSC, are equivalent in efficacy to close this artificial scratch wounding assay. This assay would become a standard procedure to release batches of manufactured composite-engineered skin.

A couple of existing engineered skins are on the market to treat chronic wound healing. These composite skin substitutes are mostly based on allogeneic primary adult skin cells or with spontaneously immortalized human keratinocyte cell lines [33]. These skin substitutes use bovine collagen I for dermal scaffold matrix that could be considered as xenogeneic raw materials [32].

To produce a personalized skin substitute, a banking of autologous cells would be very expensive, due to the requirement of repeated production and quality controls for each patient or donor. Control and qualification of each of these independent donors’ batches would add to the complexity and cost. Moreover, the use of primary adult allogenic cells could induce a heterogeneity of batch productions at the long term.

Human pluripotent stem cells such as hESCs and hiPSCs seem to be an alternative source of cells to develop production of therapeutic skin substitutes due to their ability to proliferate infinitely [34]. That property can permit to generate large banks of KER and FIB all derived from the same donor [10,34]. Obtaining all needed cells from a single donor facilitates production processes and quality controls required for clinical productions, and ensures no variability between batch productions observed to multiple donors.

In this study, we were able to obtain billions of frozen cells with multi-chamber culture devices, thus reducing container handling and thus contamination risks. These amounts of cells allow the production of an equivalent of 15,000 cm^2^ of reconstructed dermo-epidermal tissue derived from hPSCs.

## 5. Conclusions

In this study, we propose to produce a GMPc human pluripotent stem cell-derived engineered composite skin. The main advantage of using pluripotent stem cells to manufacture an engineered skin is that these cells could be amplified and differentiated *ad infinitum*. These manufacturing processes are robust and reproducible with the GMPc hESC line and could be used to produce ATMP with a huge industrial scale up perspective [35]. Finally, the hPSC cell line could be developed from specific donors to obtained haplobanks [36] or could be genetically manipulated to evade the immune system [37] in order to obtain “universal graftable” engineered composite skins avoiding tissue rejection. Altogether, this strategy could lead to a reduction in the cost in the engineered skin production in order to offer a treatment to the greatest number.

## 6. Patents

Findings from this research have been filed as part of a patent application EP 20305218.0: “Methods for preparing keratinocytes” and EP 20305214.7: Procédé de différenciation de cellules souches pluripotentes en fibroblastes de tissus conjonctifs sous-jacents d’un épithélium.

## Figures and Tables

**Figure 1 cells-11-01151-f001:**
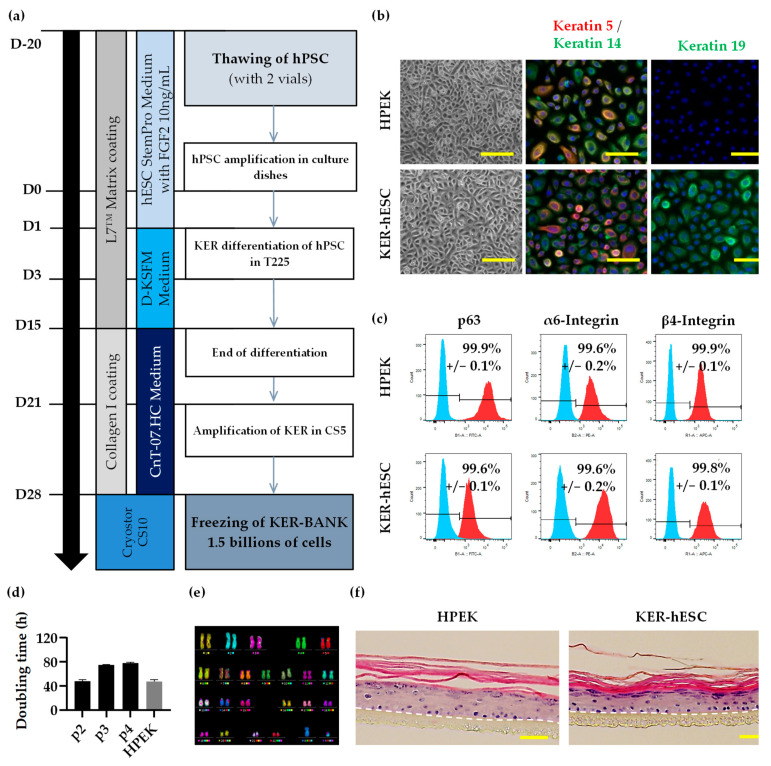
Characterization of a homogeneous and functional population of keratinocytes derived from hESC. (**a**) Schematic representation of the clinical compatible protocol design for differentiation of hPSC into keratinocytes. (**b**) Microscopic observation of HPEKp3 and KER-hESCp2 morphology (Scale bars: 200 µm) and immunocytochemistry analysis of keratin 5, keratin 14, and keratin 19 (Scale bars: 100 µm). (**c**) Flow cytometry analysis of p63, α6-Integrin, and β4-Integrin expression (in red). Staining with isotypic antibody (in blue) was performed as control. (**d**) Doubling time of KER-hESC from passage 2 (p2) until p4 and HPEKp3. (**e**) Karyotype analysis of KER-hESC (46:XY) by mFISH staining. (**f**) Eosin-Hematoxylin staining of epidermal reconstitution on polycarbonate membrane (Scale bars: 100 µm).

**Figure 2 cells-11-01151-f002:**
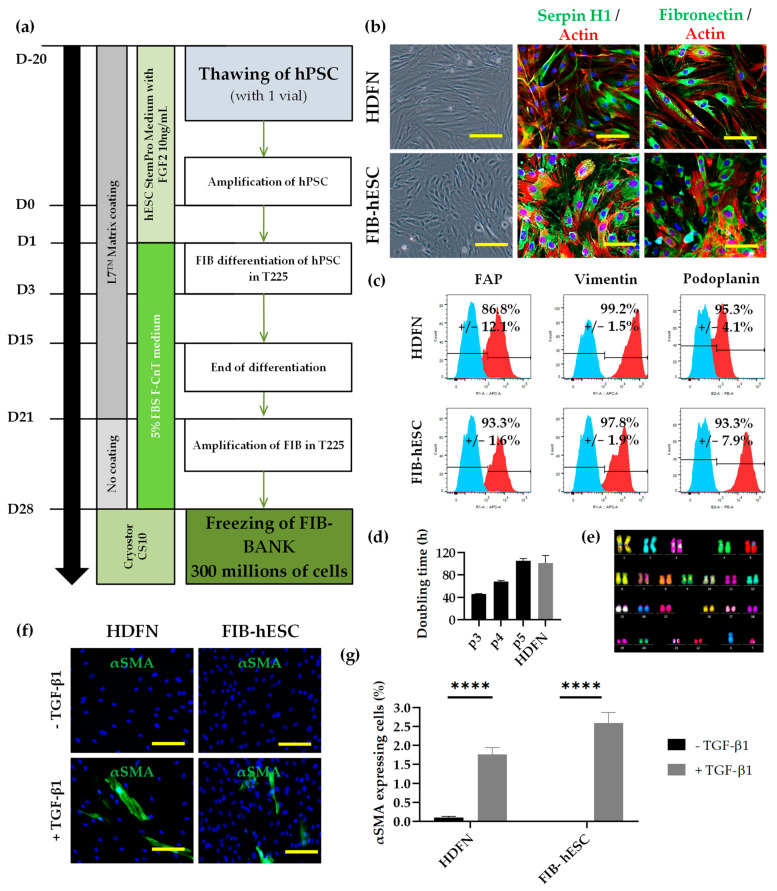
Characterization of a homogeneous and functional population of fibroblasts derived from hESC. (**a**) Schematic representation of the clinical compatible protocol design for differentiation of hPSC into fibroblasts. (**b**) Microscopic observation of HDFN and FIB-hESC morphology (Scale bars: 200 µm) and immunocytochemistry analysis of Serpin H1 and Fibronectin (Scale bars: 100 µm). The cell’s shape was observed with actin staining (in red). (**c**) Flow cytometry analysis of FAP, Vimentin, and Podoplanin (in red). Staining with isotypic antibody (in blue) was performed as control. (**d**) Doubling time of FIB-hESC from passage 3 (p3) until p5 and HDFNp4. (**e**) Karyotype analysis by mFISH staining of FIB-ESC (46:XY). (**f**,**g**) Immunocytochemistry analysis and associated quantification of αSMA in differentiated HDFN and FIB-hESC after TGF-β1 stimulation (Scale bars: 200 µm). For statistical significance: **** *p* < 0.001.

**Figure 3 cells-11-01151-f003:**
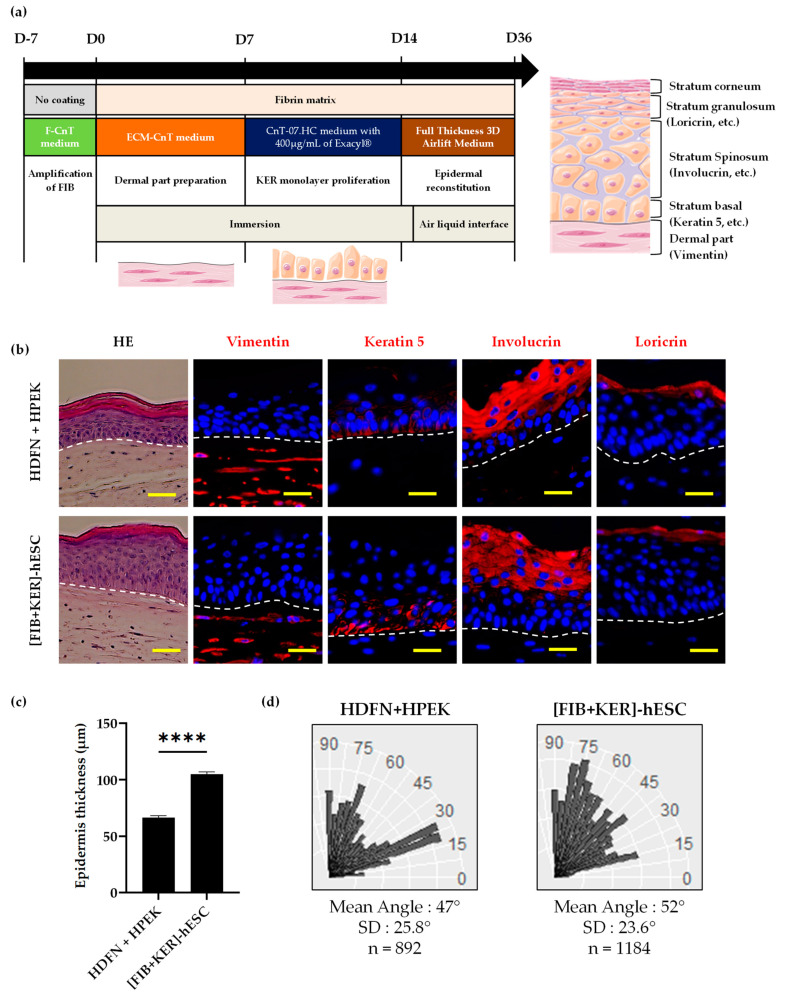
Functional characterization of composite skin substitute derived from hESC. (**a**) Schematic representation of the clinical production of dermo epidermal reconstituted tissues. Illustrated using Servier Medical Arts-SMART image bank. (**b**) Epidermal reconstitution of keratinocytes on plasma-based matrix containing fibroblasts analysis by Hematoxylin-Eosin staining (Scale bars: 100 µm) and immunohistochemistry analysis (Scale bars: 50 µm) of dermal marker vimentin, epidermal basal layer marker keratin 5, and suprabasal markers involucrin and loricrin. (**c**) Epidermis thickness measurement between the basal and beginning of the corneal layer (µm). (**d**) Distribution of basal keratinocyte nuclei according to angle versus the DEJ plan into angle categories from 0° to 90°, characterized by automated image analysis. The vertical axis represents angle values and the horizontal axis numbers of cells in the different angle categories. n corresponds to the number of analyzed nuclei. For statistical significance: **** *p* < 0.001.

**Figure 4 cells-11-01151-f004:**
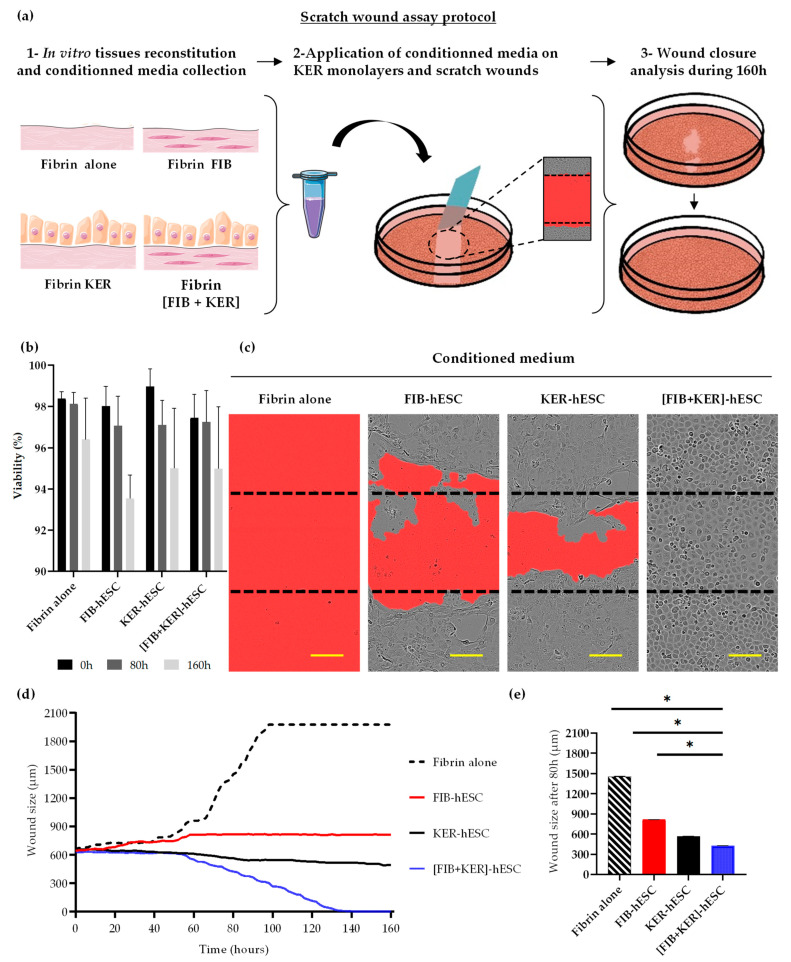
Conditioned medium from hESC-derived dermo-epidermal reconstituted tissue allows keratinocyte wound closure in vitro. (**a**) Schematic representation of wound healing in vitro protocol. After preparation of in vitro tissues (Fibrine alone, KER alone or FIB alone or [KER + FIB]), conditioned media were collected and applied on keratinocyte monolayers (at T = 0 h). Then, the monolayers were scratched using Incucyte^®^ 96-well WoundMaker Tool (Essen Bio Science Inc., Ann Arbor, MI, USA)and the wound closure was analyzed for 160 h. Illustrated using Servier Medical Arts-SMART image bank. (**b**) Viability measured on keratinocyte monolayer with Incucyte^®^ device and software (version: 20181.16628.28170, Incucyte Zoom 2018A, Essen Bio Science Inc., Ann Arbor, MI, USA). (**c**) Wound appearance 160 h after conditioned media treatments (Scale bars: 200 µm). (**d**) Monitoring of in vitro wound closures during 160 h with conditioned media. (**e**) Comparison of keratinocyte wounds size of scratch areas 80 h after the beginning of the experiment. For statistical significance, * *p* < 0.05.

## Data Availability

Not applicable.

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
