# Peer review of "Clinical Grade Human Pluripotent Stem Cell-Derived Engineered Skin Substitutes Promote Keratinocytes Wound Closure In Vitro"

_cells, 2022, doi:10.3390/cells11071151_

Round 1

Reviewer 1 Report

The study termed “Clinical grade human pluripotent stem cells derived engineered skin substitutes promote keratinocytes wound closure in vitro” proposes a Good Manufacturing Practices compliant (GMPc) human pluripotent stem cells-derived engineering composite skin.

The laboratory test using cell lines model is well justified. The manuscript is generally well written, understandable and the experiments are well described. Therefore, the engineered skin production can be tested/used in skin wound injury. Very well assessed protocol. 

Author Response

We thank the reviewers for their comments and evaluation of our manuscript.

Answers can be found in the attached file

Reviewer 2 Report

Domingues and authors studied the human pluripotent stem cells derived engineered skin substitutes promote keratinocytes wound closure in vitro. The development of engineered skin substitutes is very interesting, the research towards the development of engineered skin substitutes is important for chronic wounds. Even though the manuscript is interesting but in-vitro experiments only may not warrant publication in cells. Further in vivo experiments are required to prove the in-vitro findings.

Author Response

(The authors gave the same response as above.)

Reviewer 3 Report

In the current study Domingues et al have generated keratinocytes (KER) and fibroblasts (FIB) from human pluripotent stem cells (hPSCs) such as embryonic stem cells (hESCs) and induced pluripotent stem cells (hiPSCs). The authors have shown that the characteristics of hPSC-derived KER and FIB cells are comparable to normal human primary embryonic keratinocyte (HPEK) and normal human dermal fibroblasts neonatal (HDFN) respectively. Furthermore, a dermo-epidermal tissue was reconstructed using these hPSC-derived KER and FIB cells which showed a similar stratification to HPEK and HDFN dermo-epidermal reconstituted tissue. Finally, the authors demonstrated that the conditioned media from the dermo-epidermal tissue reconstructed from the hPSC-derived KER and FIB is capable of healing keratinocyte wound in in vitro scratch assay. The manuscript is well written, the methodology is very clear, and the data are also convincing. However, the manuscript would be more appealing if the following comments are taken care of.

Comments

  • It would be great if the wound healing assay is validated in in vivo models. It could potentially be tested in normal as well as impaired wound healing models.
  • It would also be interesting to see what factors in the conditioned media from the dermo-epidermal tissue reconstructed from the hPSC-derived KER and FIB underlie improved wound healing. It will give some mechanistic insight.

Author Response

(The authors gave the same response as above.)

Round 2

Reviewer 2 Report

The authors have sufficiently addressed the comments on why they couldn't do in vivo experiments. 

Reviewer 3 Report

The authors have sufficiently addressed the comments although they have not included the data in the revised manuscript. The manuscript can be accepted now.